# Lipid Storage and Therapy Resistance in Chronic Myeloid Leukaemia: A Novel Perspective on Targeting Metabolic Vulnerabilities

**DOI:** 10.3390/cancers17183033

**Published:** 2025-09-17

**Authors:** Molly Tolland, David M. Ross, Deborah White, Timothy P. Hughes, Ilaria S. Pagani

**Affiliations:** 1Precision Cancer Medicine Theme, Blood Cancer Program, South Australian Health & Medical Research Institute, Adelaide 5000, Australia; molly.tolland@sahmri.com (M.T.); david.ross@sa.gov.au (D.M.R.); deborah.white@sahmri.com (D.W.); tim.hughes@sahmri.com (T.P.H.); 2School of Medicine, Faculty of Health and Medical Sciences, University of Adelaide, Adelaide 5000, Australia; 3Department of Haematology and Bone Marrow Transplantation, Royal Adelaide Hospital, Adelaide 5000, Australia; 4Centre for Cancer Biology, University of South Australia, Adelaide 5000, Australia; 5Department of Haematology and Genetic Pathology, Flinders University and Medical Centre, Adelaide 5042, Australia

**Keywords:** chronic myeloid leukaemia, tyrosine kinase inhibitors, lipid, metabolism

## Abstract

Chronic Myeloid Leukaemia (CML) patients who display resistance to multiple Tyrosine Kinase Inhibitors (TKIs) remain an obstacle in the treatment of CML. Therapeutic targeting of metabolic vulnerabilities in combination with TKIs has been proposed as a potential future treatment direction. Lipid storage in the form of Lipid Droplets (LDs) is emerging as a cellular process implicated in the progression of several different cancers; however, it is relatively unexplored in CML. Here we review preclinical and clinical studies on targeting lipid storage (both directly and indirectly) in CML, evaluating reported outcomes and propose future research directions.

## 1. Introduction

Chronic Myeloid Leukaemia (CML) is a haematologic malignancy characterised by the formation of the Philadelphia chromosome [1,2], harbouring the *BCR::ABL1* fusion gene [3]. The BCR::ABL1 protein is a constitutively active tyrosine kinase, which drives oncogenic transformation [4]. The introduction of Tyrosine Kinase Inhibitors (TKIs), which specifically target BCR::ABL1, have transformed the disease from a once fatal condition to a chronic disease for most patients [5]. Patient treatment response is monitored by measuring *BCR::ABL1* transcript levels in the peripheral blood. Molecular Response (MR) milestones for CML patients are determined based on the reduction in *BCR::ABL1* mRNA and are defined as Major Molecular Response (MMR/MR3; *BCR::ABL1* ≤ 0.1% International Scale (IS), MR4 (*BCR::ABL1* ≤ 0.01% IS), and MR4.5 (*BCR::ABL1* ≤ 0.0032% IS). Achievement of MR4 or deeper is termed a Deep Molecular Response (DMR) [6].

Importantly, the response to TKIs varies among patients (Figure 1). Nearly half of all CML patients will achieve a DMR [7] and some will cease treatment entirely, without disease relapse, reaching a Treatment Free Remission (TFR) [8,9]. However, there remains patients who display a suboptimal treatment response [10] or resistance to multiple TKIs, including ponatinib [11], which is the recommended TKI following previous resistance to second generation TKIs [12]. Patients who do not respond to treatment will progress to the final, and often fatal, blast crisis stage of disease [13]. The 1-year overall survival (OS) for patients who exhibit treatment failure with ponatinib as their third-line treatment is 54% [11], highlighting the need for new treatment options. Moreover, most patients who do respond to treatment require ongoing TKI therapy to prevent disease progression and relapse arising from residual CML stem/progenitor cells that are not eradicated by TKIs [14]. This can impact patient quality of life since long-term TKI use can be associated with adverse events, such as gastrointestinal toxicity, oedema, headaches, nausea, and fatigue [15]. A particularly concerning complication associated with TKI use is cardiovascular toxicities, which occur more frequently in patients receiving nilotinib [15] and ponatinib [16]. Drug adherence is also a challenge for patients on long-term TKI therapy and a correlation between medication adherence and disease outcomes has been demonstrated [15]. Furthermore, patients who achieve TFR report improved social functioning and quality of life, highlighting the importance of enabling treatment cessation [17]. To overcome the limitations of current treatment options, novel strategies are needed to specifically target resistant cells and improve patient outcomes.

There are various mechanisms of TKI resistance with the most well characterised being ABL1 kinase domain mutations, such as the T315I mutation, *BCR::ABL1* overexpression, and altered expression of drug transporters [18]. In patients with *BCR::ABL1* resistance mutations, switching to an alternative TKI is often sufficient for disease management [6]. Additionally, there are emerging *BCR::ABL1* independent mechanisms of resistance including metabolic rewiring [19]. Metabolic reprogramming is considered a hallmark of cancer [20] due to the differing energy demands of cancer cells [21]. Alterations in lipid metabolism, specifically, are common to several cancers [22] including CML where more primitive, treatment-evading CML cells display reprogramming of lipid metabolism [23]. Such metabolic adaptations may support the survival of CML cells exposed to TKI treatment and could, therefore, be targeted using an adjuvant therapy. In this review, we focus on lipid storage, a critical branch of lipid metabolism that appears to be exploited by different types of cancer cells [24] and is a potential therapeutic target [25]. Since lipid storage is relatively unexplored in CML, here, we evaluate the limited studies available to determine where future research on this topic should be directed.

## 2. Reprogramming of Lipid Metabolism in Cancer: Emerging Insights in CML

In humans, lipids are mostly obtained from dietary sources; however, cells can also perform de novo lipogenesis to generate lipids from non-lipid precursors, including carbohydrates and glutamine. While this process mainly occurs in hepatocytes and adipocytes [26,27], rat studies show that other cell types also synthesise lipids, including kidney, lung, heart, and spleen tissue [28]. Importantly, increased expression of the gene encoding key lipogenic enzyme fatty acid synthase has been observed in CD34+ stem/progenitor cells from CML patients compared to healthy controls, indicating enhanced lipogenic activity [29]. Lipids have a range of cellular functions including membrane formation [30,31], ATP generation [32], cell signalling, apoptosis [33], and regulating ferroptosis, a form of cell death characterised by an increase in lipid peroxides [34]. Additionally, the activity of fatty acid metabolism enzyme, arachidonate 15-lipoxygenase, appears to be critical for haematopoietic stem cell maintenance [35].

Lipid metabolism is dysregulated in various cancers, including breast, prostate, and colorectal cancer, promoting cell proliferation and tumour progression (as reviewed in Broadfield et al. and Liu et al. [22,36]). These metabolic alterations can serve as markers of disease severity [37,38] and are, therefore, potential prognostic biomarkers. Such biomarkers can play an important role in guiding cancer treatment, allowing for more personalised therapy. Moreover, pharmacological targeting of lipid metabolism as an anti-cancer therapy, has shown tumour suppression capabilities in xenograft models of solid tumour cancers [39,40] and so further clinical investigation into lipid metabolism pathways as therapeutic targets is warranted.

Metabolic adaptations are one of the potential mechanisms of TKI-resistance in CML [19]. The ability of CML cells to adapt in response to and subsequently evade TKI treatment was shown in a study of transgenic mice by Qiu et. al. [41]. CML stem/progenitor cells, isolated following prolonged TKI-treatment, showed restoration of processes, including oxidative phosphorylation and tricarboxylic acid cycle activity, that were inhibited at treatment initiation [41]. In patient samples, CML stem/progenitor cells exhibited changes in lipolysis and fatty acid oxidation compared to more differentiated and thus more TKI-sensitive leukaemic cells [23]. Analysis of publicly available gene expression datasets [42,43] revealed a reduction in ferroptosis in CML patient blood or bone marrow samples compared to healthy control samples. Reduced ferroptosis was also observed in blast crisis patient samples compared to those in the chronic phase [44]. Additionally, in a study comparing good and poor responders to TKI treatment, an enrichment of gene expression associated with fatty acid metabolism was observed in *BCR::ABL1+* cells from poor responder patients [45]. These alterations suggest an adaptive metabolic plasticity in CML cells that may support their reduced TKI sensitivity and suggest that they could be targeted as a potential adjuvant therapeutic strategy. Building on the findings that lipid storage is a potential therapeutic target in solid tumours [24,46,47] and the lipid metabolism alterations observed in CML [23,41,45], we have focused on investigating the potential relevance of lipid storage, specifically, in CML.

## 3. Lipid Droplets as a Potential Therapeutic Target in CML

Intracellular lipid storage occurs primarily in the form of Lipid Droplets (LDs) (Figure 2) which consist of a neutral lipid core, composed mainly of Cholesterol Esters (CEs) and triacylglycerols surrounded by a phospholipid monolayer [48]. The LD membrane also contains membrane proteins termed Lipid Droplet Associated Proteins (LDAPs), which play a critical role in the storage and breakdown of LDs. The LD proteome has been profiled in both human and murine cells, with the number and type of LDAPs varying based on cell type and the condition of the extracellular environment [49,50]. Critical LDAPs include perilipins [51,52], G0/G1 Switch Gene 2 (G0S2) [53], and Adipocyte Triglyceride Lipase (ATGL) [54].

While adipocytes are specialised lipid storing cells in humans, LDs are found in most cell types [56], including cancer cells [25]. The structure and range of functions of LDs have been reviewed in detail [48,56,57,58]. Their main function is the supply of fatty acids under nutrient deprivation conditions, but LDs can also act as a buffer against toxic lipid peroxide species, thereby modulating susceptibility to ferroptosis [59]. These functions can, therefore, promote cell survival.

LD metabolism is regulated by multiple pathways involving LDAPs [60,61] in addition to other lipogenic and lipolytic enzymes. One critical regulatory mechanism is lipophagy [62], a selective form of autophagy that targets LDs for degradation, thereby facilitating lipid recycling and energy homeostasis under conditions of nutrient stress [62,63]. Conversely, cellular LD content can also regulate autophagy as shown by Dupont et al. [64]. The in vitro HeLa cell study showed that increased LDs following oleic acid treatment also led to increase autophagic capacity. Moreover, knockdown of the gene encoding LDAP, Patatin-like Phospholipase Domain Containing 5 (PNPLA5), inhibited LD consumption and reduced autophagic capacity [64]. Given their role in metabolic regulation and cellular stress responses, LDs are increasingly recognised as potential contributors to cancer biology [24]; however, their precise roles in the survival of different cancer cells remains to be fully defined.

LDs appear to be implicated in the disease progression of several cancers [24,25] including colon [65], breast [66], and brain cancer [67]. In breast cancer, LDs promote cell survival during nutrient deprivation and lipotoxic stress [66], and in glioblastoma patients, LD accumulation was found to inversely correlate with overall survival [67]. Moreover, targeting LDAPs, such as Diacylglycerol-Acyltransferase 1 (DGAT1) and Hypoxia Inducible Lipid Droplet Associated (HILPDA) can block LD formation and suppresses tumour growth as shown in xenograft models of glioblastoma [46] and pancreatic cancer [47], respectively. Studies focused on LDs in CML are limited; however, a recent study found that increased uptake of free fatty acids and LD accumulation suppressed BCR::ABL1 protein expression in K562 and KCL22 cells [68]. Under hypoxic and glucose starved conditions, which model the CML stem/progenitor cell environment, the cell lines exhibited a glutamine-dependent increase in CD36 receptor expression leading to a subsequent increase in fatty acid uptake and LDs. Importantly, increased fatty acid accumulation suppressed BCR::ABL1 protein expression, which is known to be associated with cell quiescence and, therefore, reduced TKI sensitivity in CML progenitor cells. Based on these findings the authors suggested that inhibiting fatty acid uptake with CD36 inhibitor, sulfo-N-succinimidyl oleate, may disrupt stem/progenitor cell maintenance in CML [68].

Given the evidence of dysregulated lipid metabolism in CML [23,41] and the potentially pathogenic role of LDs in cancer, more generally [65,66,67] studying LD dynamics in CML could reveal novel targetable vulnerabilities. Here, we investigate three potential approaches for targeting LDs in CML: inhibition of lipid biosynthesis pathways, regulation of LDAPs, and modulation of autophagy. We focus on clinical studies and critically evaluate their outcomes, as summarised in Table 1.

## 4. Targeting Lipid Storage in CML as an Adjuvant Therapy

### 4.1. Inhibition of Lipid Biosynthesis Pathways: Targeting Cholesterol Storage in CML

Cholesterol is a critical component of cell membranes, and its homeostasis is tightly managed, with excess cholesterol stored in the form of Cholesterol Esters (CEs) within LDs [76]. Due to the necessary role of cholesterol for cellular function, it has the potential to be utilized by cancer cells to promote cell proliferation and survival. Bandyopadhyay et al. investigated CE depletion as a method to overcome imatinib resistance in *BCR::ABL1*+ K562 cells [77]. K562 cells displayed a ~40% increase in CE as a percentage of total lipids compared to each of the *BCR::ABL1-* leukaemic cell lines tested. Furthermore, Ba/F3 cells overexpressing *BCR::ABL1* demonstrated increased LD accumulation compared to empty vector cells. The authors subsequently concluded that BCR::ABL1 signalling was sufficient to promote cholesterol esters and LD accumulation. An imatinib resistant K562 cell line was also generated by culturing cells with Fibroblast Growth Factor 2 and 1 µM imatinib for ~4 weeks. These cells were subsequently re-sensitised to imatinib through combination treatment with the cholesterol esterification inhibitor, avasimibe. Importantly, imatinib and avasimibe displayed synergy in suppressing imatinib resistant K562 xenograft tumour growth [77]. Avasimibe, which inhibits Acyl Co-Enzyme A Cholesterol Transferase (ACAT) (Figure 3), was previously evaluated in clinical trials for the treatment of human coronary atherosclerosis but was found to be ineffective. Despite this, avasimibe was well-tolerated and so could be tested in clinical trials for other diseases [78]. The safety of avasimibe and ability to re-sensitise TKI resistant *BCR::ABL1+* cell lines to TKI treatment, indicates that targeting cholesterol storage may serve as an effective combination treatment strategy in TKI resistant CML patients.

Targeting cholesterol synthesis (Figure 3) in CML patients, through treatment with statins which inhibit 3-Hydroxy-3-Methyl-Glutaryl-Coenzyme A (HMG-CoA) reductase, has also been investigated as a potential treatment option. Initially, statin use in CML was recommended for patients who experienced hypercholesterolemia in response to nilotinib treatment, to reduce their Low-Density Lipoprotein (LDL)-cholesterol levels [79]. However, more recent findings suggest that statins may be effective in increasing TKI sensitivity. A synergistic anti-proliferative and enhanced pro-apoptotic effect has been observed between simvastatin and nilotinib in *BCR::ABL1+* cell lines and primary patient samples [80]. Interestingly, both atorvastatin [81] and simvastatin [82] monotherapy, reduced cell proliferation in *BCR::ABL1+* K562 cells through inducing cell cycle arrest. Furthermore, simvastatin treatment alone led to an impairment of K562 xenograft tumour growth in nude mice, however a TKI-combination treatment was not tested [82]. While these studies highlight a potential therapeutic benefit of statins in CML, the doses used in the monotherapy treatments were well above the clinical dose range.

Clinical investigations of statin and TKI combination treatment in CML are limited and conflicting findings have been reported. To date, no randomised clinical trials have been conducted to determine the clinical efficacy of statin-TKI combination therapy and the most relevant data reported are the outcomes of retrospective studies. Clinical outcomes of imatinib monotherapy and imatinib–statin combination therapy were obtained from 408 CP-CML patients, with a median follow up of 77 months [69]. The imatinib–statin group made up 21.3% of patients and was defined as those undergoing statin (either atorvastatin, simvastatin, pravastatin, or fluvastatin) therapy for hypercholesterolaemia at the time of imatinib initiation. The remaining patients were treated with imatinib only. The imatinib–statin combination group achieved a non-significant increase in the incidence of MMR at 3 years (77.3% compared to 62.5%) and a significant (*p* = 0.001) increase in the incidence of DMR (defined here as MR4.5) at 5 years (55.8% compared to 41.0%) compared to the imatinib group. After propensity score matching to control for potential interactions between statin use and other confounding clinical factors, the DMR rates at 5 years remained higher in the imatinib–statin group compared to the imatinib group (56.8% compared to 47.0%, *p* = 0.019). Furthermore, statin use was identified as an independent factor for achieving DMR [69]. While the benefit of the imatinib–statin combination appears marginal, increasing the incidence of DMR may allow more patients to trial TFR since sustained DMR is an eligibility requirement for attempting TKI cessation [6]. Achieving a TFR is an important treatment goal for some patients and has been demonstrated to drastically improve quality of life [17].

Analysis of primary bone marrow CD34+ stem/progenitor cells collected from CML patients (n = 2), and healthy control patients (n = 1) was also performed [69]. In vitro treatment with 0.6 µM imatinib and 1.5 µM rosuvastatin in combination reduced cell viability compared to the single treatments; however, rosuvastatin alone induced cell death (<20% cell viability) in both healthy control and CML CD34+ stem/progenitor cells [69]. While this appears to be a drug toxicity response, 1.5 µM is well above the clinically relevant C_max_ for rosuvastatin [83]. Future studies should test more clinically relevant rosuvastatin concentrations to determine whether this drug combination would be effective in the clinical setting and achieve more targeted CML cell death.

In another retrospective analysis of 40 CP-CML patients, the effect of statin co-treatment on imatinib response was evaluated [70]. Nineteen patients were placed in the ‘statin group’ based on receiving statins for hypercholesterolemia prior to and throughout imatinib treatment. It is important to note, however, that one of the ‘statin group’ patients was in fact treated with fenofibrate which does not fall within the statin class. The remaining statin treated patients received one of either atorvastatin, pravastatin, simvastatin, or rosuvastatin. Lipid profiles were reported at initiation of imatinib and after 3 (n = 35 patients) and 12 months treatment (n = 37 patients). The addition of imatinib to statin treatment led to a significant reduction in total cholesterol, triacylglycerol, LDL, and non-High-Density Lipoprotein (non-HDL), and an increase in HDL. These changes were sustained at 12 months of treatment apart from total cholesterol and LDL which restored to the levels seen with statin treatment alone. Similar lipid lowering effects were seen in the no-statin group excluding total cholesterol which remained unchanged. Importantly, no significant differences were observed between the two groups for the time to *BCR::ABL1* reduction (to *BCR::ABL1* <1%, <0.1%, <0.01% and undetectable) and leukocyte normalisation.

The authors concluded that statins did not alter the sensitivity of CML cells to imatinib, suggesting that there is no benefit for statin–imatinib co-treatment in CML [70]. These findings differ from those of Jang et al. [69] who reported improved DMR rates with the addition of statins to imatinib treatment. Both studies controlled for clinical covariates; through propensity score matching in Jang et al. and baseline characteristic comparison in Ellis et al. [70], but differed substantially in cohort size which may explain the discrepancy in outcomes. Despite these conflicting results, the outcome of Jang et al. [69] is a promising finding and warrants further investigation in to the potential for statins to enhance the TKI response in CML.

Future investigations of statin-TKI treatment combinations should analyse larger cohorts and include patients treated with second and third-line TKIs. Studying the outcomes of statin treatment with alternative TKIs is a necessary continuation of these clinical studies since nilotinib has shown a differential effect on patient serum lipid profiles, specifically total cholesterol and LDL-cholesterol levels, compared to imatinib [84]. Additionally, in vivo studies employing clinically relevant doses will be necessary to identify potential synergistic mechanisms between the two drugs.

### 4.2. Targeting LD-Associated Proteins via PPARγ

LDAPs are regulators of LD metabolism and their expression and activity represent potential therapeutic targets in CML. Peroxisome proliferator-activated receptors (PPARs) are a group of nuclear hormone receptor transcription factors that regulate numerous signalling pathways [85] including LD metabolism through mediating the expression of several genes encoding LDAPs including perilipins, HILPDA, ATGL (encoded by *PNPLA2*) [86], and G0S2 [87]; however, expression can vary based on cell type [86]. PPARγ, specifically, is expressed in various tissues including adipocytes, hepatocytes [88,89], and haematopoietic progenitor cells [90]. The transcriptional activity of PPARγ is shown in Figure 4. Its role in haematopoiesis was first noted when investigating the human immunodeficiency virus negative factor protein, which disrupts hematopoietic progenitor expansion through downregulation of STAT5A/B via PPARγ signalling. PPARγ agonists mimic this pathway, similarly reducing STAT5A/B expression and consequently inhibiting haematopoietic expansion [91].

The LDAP G0S2 is an ATGL inhibitor thereby preventing LD degradation through suppression of lipolysis [53]. In CML patients, *G0S2* expression was decreased in the CD34+ stem/progenitor cell compartment in blast phase CML (n = 5) compared to chronic phase patients (n = 6) [92]. An even greater decrease in expression was observed in CD34+ cells from TKI-resistant CML patients (n = 3) exhibiting *BCR::ABL1*-independent resistance. Additionally, in CD34+ cells from n = 35 CP-CML patients, lower *G0S2* expression was associated with reduced 10-year overall survival. Functional studies in K562 cells lentivirally transduced for *G0S2* overexpression or knockdown revealed that *G0S2* loss alters glycerophospholipid metabolism, autophagy, and ferroptosis. Interestingly, *ATGL* (*PNPLA2*) knockdown in K562 cells did not replicate the effects of ectopic *G0S2* expression on gene signalling, suggesting that G0S2 has ATGL-independent functions. This observation is important because a decrease in *G0S2* expression being associated with disease progression in CML would, in theory, suggest a reliance on free fatty acids generated by the ATGL-mediated lipolysis of LD triacylglycerols. However, the ATGL-independent function of G0S2 indicated by these results suggests that this alteration does not support cell survival through increased lipolysis. Despite this, lipidomics analysis revealed that *G0S2* loss in K562 cells disrupts lipid metabolism and the authors proposed that these lipid profile changes, over the course of disease progression, reduce TKI sensitivity in CML [92]. The observed association between reduced *G0S2* expression and advanced disease progression underscores its potential relevance as a prognostic biomarker in CML. Importantly, the observed ATGL-independent activity of G0S2 highlights potential broader regulatory functions of G0S2 on lipid metabolism in CML that should be investigated further.

While the LD regulatory effects of PPARγ, specifically, have not been explored in CML, several clinical studies have examined whether the therapeutic activation of PPARγ could improve the molecular response in CML [71,72,73,74]. Each of these studies evaluated the efficacy of TKI treatment in combination with the PPARγ agonist pioglitazone, an insulin sensitising drug used for the treatment of type II diabetes [93].

In one study, pioglitazone was added to the treatment of three CML patients, two of which also had type II diabetes, who had not reached a complete molecular response (CMR) (defined as undetectable *BCR::ABL1* by RT-qPCR) after 4–6 years on imatinib [71]. Each patient received pioglitazone for different treatment lengths with a maximum treatment period of 28 months. One patient stopped imatinib treatment 6 months into the study while the others remained on imatinib. CMR was achieved in each of the patients 10, 12, and 6 months after the introduction of pioglitazone [71].

This initial observation was followed by the Actos + Imatinib (ACTIM) study [72], which enrolled 24 CP-CML patients who had received imatinib for a median of 73 months but had not achieved MR4.5. At the time of study initiation, 58% of patients were in MMR without achieving MR4 and 42% of patients were in MR4 without achieving MR4.5. Patients then received 30–45 mg/day of pioglitazone for a period of 1.9 to 15.5 months. Pioglitazone, in addition to imatinib, increased the cumulative incidence of MR4.5 over 12 months to 56% of patients compared to 23% in a historical cohort with similar characteristics receiving imatinib only. The increased cumulative incidence of MR4.5 is potentially due to the lengthened period of imatinib treatment [94] as this has been observed in long-term clinical trials (23% of imatinib treated patients achieved MR4.5 by 4 years and 31% of patients by 5 years) [95]. However, the historical cohort comparison suggests that the addition of pioglitazone to imatinib contributed to the deeper response observed. *STAT5* expression was also reduced in CD34+ stem/progenitor cells from patients treated with pioglitazone. The authors, therefore, suggested that the deeper response to the pioglitazone–imatinib combination was through the pioglitazone induced decrease in *STAT5* expression. While PPARγ is a critical regulator of lipid metabolism, it is involved in multiple cellular pathways and so cell death induced by PPARγ agonists could be through one of multiple mechanisms. Future study of CML patient CD34+ stem/progenitor cell lipid profiles may highlight whether this specific regulatory function of PPARγ is involved in the CML TKI-response.

A separate pioglitazone study was conducted involving 31 patients on either imatinib, nilotinib, or dasatinib who had previously displayed a suboptimal molecular response but were negative for *BCR::ABL1* mutations [73]. Patients received 30 mg of pioglitazone daily for the entire study duration with the follow up period ranging from 59–1117 days. TKI treatment + pioglitazone led to a significant reduction in *BCR::ABL1* expression (1-log in 89.7% of study participants) over time for a median duration of 602 days. A total of 48.3% of patients achieved MMR and 19.3% achieved DMR at the time data was censored. While prolonged TKI treatment could also explain the deeper molecular response observed, the consistency of these results with the ACTIM [72] study suggest a potential additive effect of pioglitazone.

Despite the potential of pioglitazones to increase the incidence of MR4.5 [72], this does not appear to translate into increased rates of TFR as demonstrated by Pagnano et al. in the EDI-PIO study [74]. CP-CML patients who had maintained MR4.5 for at least three years received 30 mg/day pioglitazone in addition to imatinib for 3 months before discontinuing both drugs. The TFR incidence at 19 months and was 60% (CI 95%: 42–78%), which is comparable to TFR incidence after imatinib treatment alone [8,9,74], indicating that pioglitazone may not exert the same effects in deeper responders as in patients who have not yet achieved MR4.5. It is also possible that 3 months of pioglitazone treatment is an insufficient duration for enhancing patients’ TKI response. Treatment duration is an important consideration and may be responsible for the discrepancy in results between the pioglitazone and TKI combination clinical studies.

In vitro studies revealed that PPARγ agonists reduce BCR::ABL1 kinase inhibition by imatinib in *BCR::ABL1+* cell lines and OCT-1 activity in diagnosis CML MNCs [96]. However, reduced BCR::ABL1 inhibition did not translate to a reduction in sensitivity to imatinib-induced cell death. Since PPARγ activation also decreases *STAT5* expression in CML stem/progenitor cells, it was hypothesised that a concurrent reduction in *STAT5* could counteract the effect of decreased OCT-1 activity. These findings do not indicate any benefit of adding PPARγ agonists to TKI treatment in CML.

The conflicting results of these studies may be attributed to differences in the treatment timepoints. The samples analysed in the OCT-1 in vitro study [96] were obtained at diagnosis whereas the clinical study patients had all received TKI treatment for a minimum of three months prior to study entry [71,72,73,74]. With these details considered, the interaction between PPARγ signalling and BCR::ABL1 inhibition may vary depending on patient treatment history.

The group of patients most likely to benefit from the use of PPARγ agonists are those who have sustained MMR or MR4 but are struggling to reach MR4.5 [72]. The current data does not support using PPARγ agonists to increase the incidence of TFR in patients already at MR4.5. Further clinical investigation is needed to determine whether PPARγ agonist-induced MR4.5 is more stable or more likely to lead to a successful TFR. Future studies should also explore longer therapy durations since this may impact patient response to PPARγ and TKI combination treatment.

Since PPARγ has a broad range of transcriptional targets, the enhanced molecular response in pioglitazone treated CML patients reported in Rousselot et al. [72] and Yanamandra et al. [73], may not specifically result from changes in LD regulation. However, in an extension of the EDI-PIO study [71], lipid remodelling was observed in patients following pioglitazone treatment [97]. Plasma samples from 10 patients before, during, and after pioglitazone and imatinib combination treatment, were analysed by mass spectrometry. The addition of pioglitazone led to a decrease in free fatty acid abundance compared to imatinib treatment alone. When the pioglitazone + imatinib samples were compared to treatment discontinuation samples, diacylglycerol, triacylglycerol, plasmalogen, and CE levels were reduced and the authors suggested that pioglitazone treatment enhanced cytosolic lipolysis [97]. However, cellular lipidomics analysis would be needed to confirm that cytosolic lipolysis is driving this change in plasma lipid content.

Based on these results, PPARγ agonists do appear to alter CML patient lipid profiles; however, PPARγ regulates multiple cellular processes and so it remains unclear if the regulation of LDAPS, specifically, promotes survival in TKI-treated CML cells.

### 4.3. Targeting Autophagic Regulation of LDs

Autophagy is a critical process that is tightly regulated by cellular energy and nutrient sensing pathways [98] and the selective autophagic breakdown of LDs, known as lipophagy (Figure 5), is a novel area of cancer research. There are currently no drugs that target lipophagy specifically, but several autophagy activating and inhibiting drugs are available such as rapamycin [99] and chloroquine (CQ) [98], respectively. Targeting autophagy in cancer is complex since the role of autophagy and more specifically, lipophagy, appears to be context dependent and both stimulation and inhibition of these processes have been proposed as anti-cancer strategies [98]. For example, active lipophagy supports the survival of glioblastoma cells through providing free fatty acids for β-oxidation [100] whereas impaired lipophagy appears to facilitate disease progression in clear cell renal cell carcinoma (ccRCC) [101].

Another recent study in ccRCC demonstrated the ability of lipophagy to promote metastatic transformation [103]. While normal cells undergo a form of cell death termed anoikis following detachment from the extracellular matrix (ECM), metastatic cells are resistant to this process. The authors generated ‘high-metastatic’ and ‘low-metastatic’ potential ccRCC cell lines, selecting cells based on their migration and invasion capacity as well as expression of ECM proteins. Cells with ‘high-metastatic’ potential showed an increase in LDs, free fatty acids, and an enrichment for genes related to apoptosis, autophagy, and lipid metabolism. Increased co-localisation of the autophagy marker LC3 and LDs was also observed compared to the low-metastatic potential cells. Based on these results, the authors suggested that these cells are dependent on lipophagy for the release of free fatty acids involved in activating the oncogenic tyrosine kinase, Src, and subsequently promoting metastasis. Importantly, this dependency created a metabolic vulnerability and CQ treatment induced cell death in the ‘high-metastatic’ potential ccRCC cells.

While lipophagy has not been directly studied in CML, general autophagy modulation has been explored in the clinical setting [75]. Imatinib is a known autophagy inducer in CML cells and imatinib and CQ combination treatment increases cell death in K562 cells compared to imatinib alone [104]. Based on this, a phase II clinical trial evaluated the efficacy of hydroxychloroquine (HCQ), a CQ derivative, and imatinib combination treatment compared to imatinib alone [75]. The cohort consisted of 50 CP-CML patients who had received imatinib treatment for >12 months and achieved at least a Major Cytogenic Response (MCyR) but were still *BCR::ABL1*+ by qPCR analysis. Patients were randomised to receive either imatinib alone or imatinib + HCQ (400 mg twice daily) for up to 12 4-week cycles. Importantly six patients received dose reductions of HCQ due to intolerance.

Treatment success was defined as patients achieving > 0.5-log reduction in *BCR::ABL1*%. At 12 months, no difference in treatment success was observed between the two groups. At 24 months, the success rate was 20.8% higher in the imatinib + HCQ group compared to the imatinib only group; however, this did not reach statistical significance. Notably, in a post-hoc analysis, patients with a higher *BCR::ABL1/ABL1* ratio at trial initiation had a 34.6% higher success rate when treated with imatinib + HCQ. However, this result must be interpreted with caution due to small patient numbers within these subgroups. An issue identified in this trial was the insufficient plasma concentrations of HCQ achieved, likely reducing the extent of autophagy inhibition and ability to sufficiently evaluate the combined effect of imatinib and HCQ. For this reason, the authors suggested that, in the future, more potent autophagy inhibitors should be tested in combination with TKIs in CML patients.

The findings of this study do not demonstrate synergy between imatinib and HCQ. Future research with a larger group of CML patients with high *BCR::ABL1:ABL* ratios would determine whether there is any benefit of HCQ addition in these patients specifically. Lipophagy specific inhibitors, as opposed to general autophagy inhibitors, would be an important future development yielding potentially different treatment outcomes. If such therapeutics become available, they should be tested in cell lines and xenograft models to determine whether they exhibit synergy with TKIs and to compare their potency with HCQ.

## 5. Conclusions and Future Directions

In CML, the focus is now on increasing the incidence of TFR [105], allowing a greater proportion of patients to safely discontinue therapy. Additionally, multi-TKI resistance remains a clinical problem and alternative treatment options are needed. Preclinical studies have demonstrated that dysregulated lipid storage is implicated in treatment efficacy. However, the mechanisms by which these alterations may support TKI resistance are unclear. Future in vitro studies would benefit from analysing different CML cell populations and samples from larger cohorts at different disease stages. This would facilitate identification of patients who may benefit the most from a treatment intervention targeting lipid storage. Parallel to therapeutic development, these studies present potential lipid biomarkers that may predict treatment response and require further investigation.

Clinical studies of therapeutics that target lipid storage-associated pathways have shown inconsistent or minimal benefit to patient outcome. Importantly, in each of the clinical studies reviewed here, only one included patients treated with TKIs other than imatinib (nilotinib and dasatinib). Future inclusion of patients receiving second- and third-generation TKIs will produce findings that are more clinically relevant to a greater percentage of CML patients. Treatment with alternative TKIs, including asciminib, an allosteric TKI, may result in differential synergism with lipid storage targeting drugs compared to the outcomes seen in imatinib treated patients. While the benefits of targeting lipid storage in CML appear limited, further research focusing on this pathway is still warranted since lipid metabolism is an extensive process with a vast range of therapeutic targets.

There are major benefits to repurposing drugs approved for use in other diseases because of their established safety profile. Promising synergies can be recognised in cases where patients with comorbidities are taking these drugs in combination with TKIs. This is particularly relevant for common drug groups such as proton pump inhibitors and cholesterol lowering drugs. While randomised clinical trials are necessary to determine treatment efficacy, retrospective analyses are a valuable method for deciding future research directions. To more efficiently identify potential adjuvant candidates, future studies would benefit from employing drug screening libraries for lipid metabolism targeting drugs. However, a focus on the development of new therapeutics that target lipid storage to achieve a more targeted response in CML patients may be warranted. The continued advancements of high throughput technologies, such as spatial metabolomics, will also facilitate future research of complex lipid systems. Finally, due to the complicated nature of lipid metabolism and role of the extracellular environment, co-culture systems and in vivo xenograft models are needed to investigate whether lipid metabolism is implicated in promoting TKI resistance and if targeting these pathways shows therapeutic potential. A deeper understanding of metabolic alterations in CML has the potential to uncover actionable vulnerabilities and inform the development of novel adjuvant therapies to overcome TKI resistance and improve patient outcomes.

## Figures and Tables

**Figure 1 cancers-17-03033-f001:**
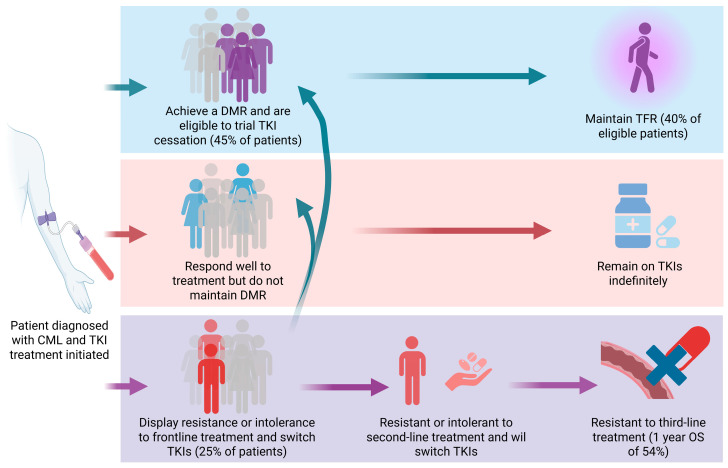
CML patient outcomes following TKI treatment. CML patients vary in their response to TKI treatment. While some patients respond optimally, achieving a DMR and potentially TFR, many patients remain on TKI therapy indefinitely. Patients who display multi-TKI resistance, including resistance to third-line TKI treatment, exhibit a worsened prognosis. Figure created using BioRender.com.

**Figure 2 cancers-17-03033-f002:**
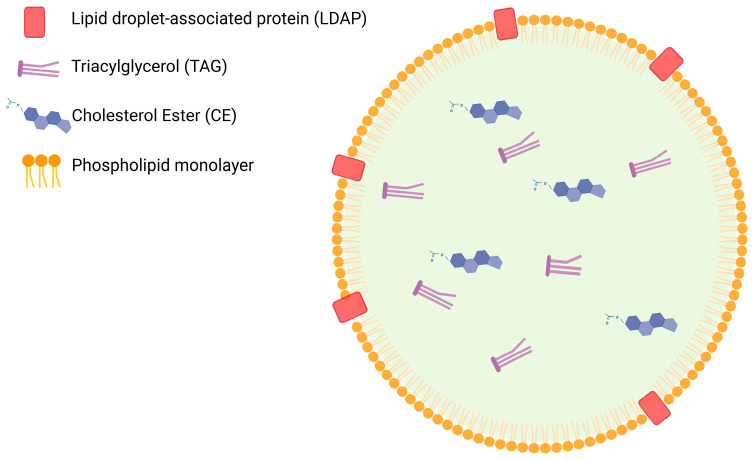
Simplified lipid droplet structure. The Lipid Droplet (LD) core is made up of neutral lipids, mostly Triacylglycerols (TAGs) and Cholesterol Esters (CEs). LDs are enclosed by a phospholipid monolayer membrane and membrane proteins termed Lipid Droplet Associated Proteins (LDAPs). This diagram was adapted from Guo et al. [55]. Figure created using BioRender.com.

**Figure 3 cancers-17-03033-f003:**
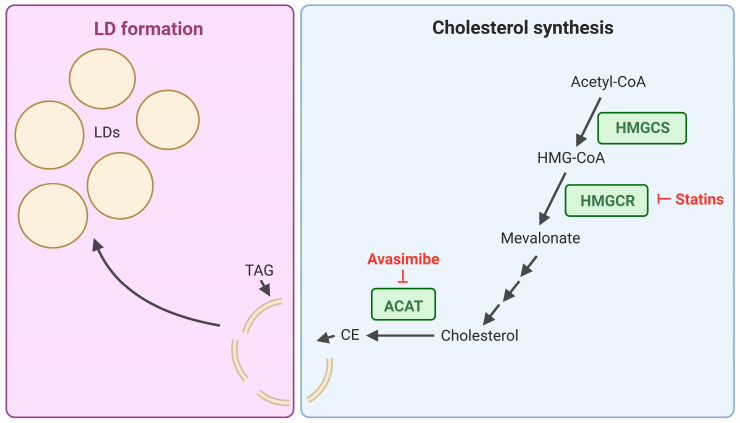
Cholesterol synthesis and incorporation into LDs. A simplified diagram of the cholesterol synthesis pathway and the incorporation of cholesterol esters (CEs) into lipid droplets. Acetyl-CoA is converted to 3-Hydroxy-3-methyl-glutaryl-coenzyme A (HMG-CoA) by HMG-CoA Synthase (HMGCS) followed by a series of enzymatic reactions involving HMG-CoA Reductase (HMGCR) to generate cholesterol. Cholesterol esterification is catalysed by Acyl Co-Enzyme A Transferase (ACAT). CEs and Triacylglycerols (TAGs) are incorporated into Lipid Droplet (LD) cores. Statins act by inhibiting HMGCR and avasimibe inhibits ACAT. Figure created using BioRender.com.

**Figure 4 cancers-17-03033-f004:**
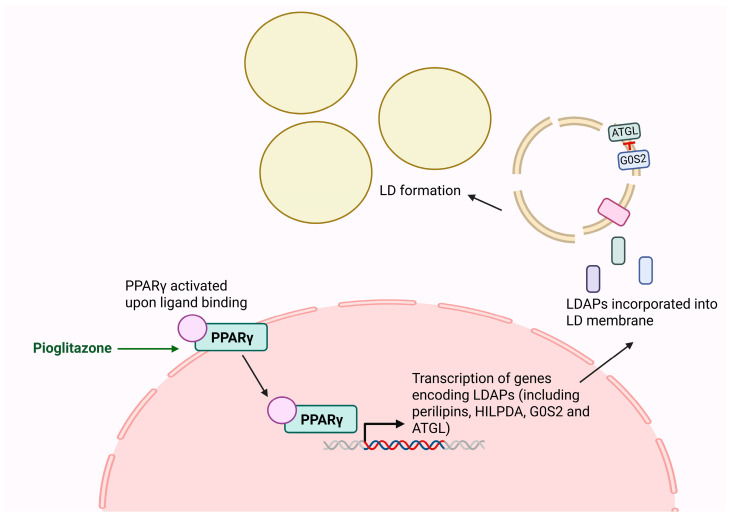
PPARγ transcriptional regulation of LDAPs. Peroxisome Proliferator-Activated Receptor γ (PPARγ) is a transcription factor that activates the expression of multiple genes encoding Lipid Droplet Associated Proteins (LDAPs) including Hypoxia Inducible Lipid Droplet Associated (HILPDA) and G0/G1 Switch 2 (G0S2) and Adipocyte Triglyceride Lipase (ATGL). These LDAPs are critical functional components of LD membranes. ATGL is a lipolytic enzyme inhibited by G0S2. Figure created using BioRender.com.

**Figure 5 cancers-17-03033-f005:**
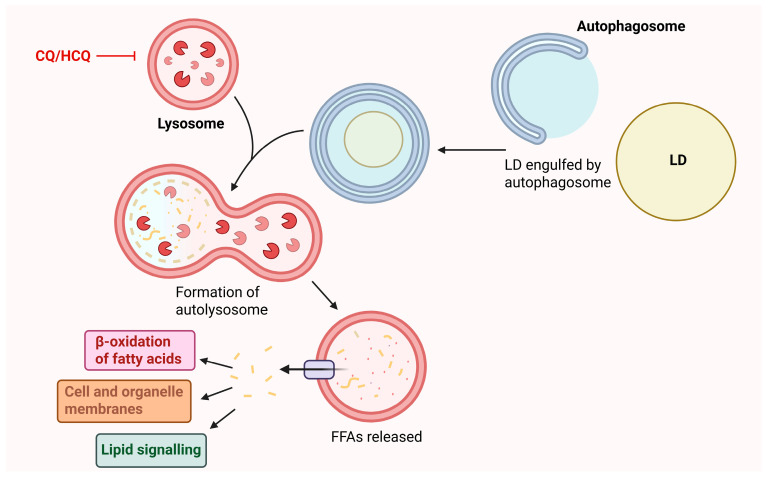
Lipophagy pathway. The autophagic breakdown of LDs, lipophagy, leads to the release of Free Fatty Acids (FFAs). FFAs can undergo β-oxidation for ATP production or be recycled to form the lipid components of cell and organelle membranes or lipid signalling molecules. Autophagy is tightly regulated by nutrient and energy sensing pathways. Chloroquine (CQ)/hydroxychloroquine (HCQ) inhibits autophagy through accumulating in the lysosome and raising the pH, rendering the lysosomal enzymes inactive. This diagram was adapted from Liu et al. [102]. Figure created using BioRender.com.

**Table 1 cancers-17-03033-t001:** Therapeutics clinically studied in CML with links to Lipid Droplet (LD) metabolism.

Drug Class	Drugs	Associated LD Pathway	No. of Patients	Patient Characteristics	Trial/Study Type	Outcome	Reference
HMG-CoA Reductase inhibitor	Atorvastatin, simvastatin, pravastatin, fluvastatin	Mevalonate pathway; Cholesterol synthesis	408; n = 88 imatinib + statin, n = 320 imatinib only	Chronic phase; median age 52 y; statin group undergoing statin therapy prior to initiation of imatinib and continued for at least 3 years alongside imatinib; 84 pairs selected for propensity score matching.	Retrospective study	Non-significant increase in MMR in the imatinib + statin group at 3 years (77.3% vs. 62.5%) compared to imatinib only group; Significant increase in DMR in the imatinib + statin at 5 years (55.8% vs. 41%) compared to imatinib only group.	[69]
HMG-CoA Reductase inhibitor; PPARα agonist	Atorvastatin, simvastatin, pravastatin, rosuvastatin; fenofibrate	Cholesterol synthesis: lipolysis	40; n = 19 imatinib + statin, n = 21 imatinib only	Chronic phase; median age 66 y; statin group undergoing statin therapy prior to and throughout imatinib treatment; comparable baseline characteristics of patients in each group.	Retrospective study	No difference in time to BCR::ABL1 reduction to BCR::ABL1 < 1%, <0.1%, <0.01% and undetectable; no difference in time taken to WBC normalisation.	[70]
PPARγ agonist	Pioglitazone	Transcription of LDAPs	3; n = 3 imatinib + pioglitazone	CMR not achieved after 4–6 years on imatinib; age range of 62–67 y; n = 2 patients with Type II diabetes.	Case study	CMR was achieved in each patient within 12 months following pioglitazone introduction.	[71]
PPARγ agonist	Pioglitazone	Transcription of LDAPs	24; n = 24 imatinib + pioglitazone	Chronic phase; median age 61 y; MR4.5 not achieved after median 73 months on imatinib.	Clinical trial: 2009-011675-79	Imatinib + pioglitazone increased cumulative incidence of MR4.5 to 56% over 12 months (compared to 23% in historical cohort treated with imatinib only).	[72]
PPARγ agonist	Pioglitazone	Transcription of LDAPs	31; n = 31 TKI (imatinib, nilotinib, dasatinib) + pioglitazone	Failure to achieve MMR after 12–15 months on TKI but *BCR::ABL1* mutation negative; median age 54 y.	Clinical trial	TKI + pioglitazone led to a significant reduction in *BCR::ABL1* expression (1-log in 87% of patients for a median duration of 602 days); At time of censoring data 48.3% achieved MMR and 19.3% achieved DMR; during follow up, disease progressed in 38% of patients.	[73]
PPARγ agonist	Pioglitazone	Transcription of LDAPs	32; n = 32 imatinib + pioglitazone (1 patient lost to centre transfer)	Chronic phase; median age 54 y, maintained MR4.5 for ≥3 y but have not achieved TFR.	Clinical trial: NCT02852486	TFR incidence at 19 months was 60%, comparable to TFR incidence after imatinib alone.	[74]
Lysomotropic agent	Hydroxychloroquine (HCQ)	Autophagy	62 (12 lost to follow up/consent withdrawal, physician intervention; n = 25 imatinib + HCQ, n = 25 imatinib only	Chronic phase; median age imatinib + HCQ group 50 y and imatinib only group 49.5 y; patients achieved MMR but still *BCR::ABL1+* after 12 months on imatinib.	Clinical trial: NCT01227135	No difference in percentage of patients achieving >0.5-log reduction in *BCR::ABL1* at 12 months; 20.8% higher rate of achieving 0.5-log reduction in *BCR::ABL1* in the imatinib + HCQ group compared to imatinib only group (did not reach statistical significance).	[75]

HMG-CoA reductase = 3-Hydroxy-3-methyl-glutaryl-coenzyme A reductase. PPARγ = Peroxisome proliferator activated receptor γ.

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
