# Peer review of "Lipid Storage and Therapy Resistance in Chronic Myeloid Leukaemia: A Novel Perspective on Targeting Metabolic Vulnerabilities"

_cancers, 2025, doi:10.3390/cancers17183033_

Round 1
Reviewer 1 Report
Comments and Suggestions for Authors
The manuscript concerns alterations of lipid contents and metabolism in chronic myeloid leukemia (CML) and effects of chemotherapy on this metabolic compartment. Available literature is relative scarce, and, in current review, the focus is made on the intracellular lipid accumulation as a factor of drug resistance in some patients who failed to respond to the TKI therapy. Of course, this therapeutic approach may be potentially useful, especially, seeking for repurposing of some drugs affecting lipid metabolism, for supplementary cancer treatment in TKI-resistant cases.
Remarks
Lines 81-84. Before discussing the data on lipid storage as a drug-resistance factor, one should mention in more details some other well-known mechanisms of TKI resistance, i.e., T355I and other mutations of BCR/ABL which make the cells insensitive to TK inhibition.
Line 132. Section 3 lipid droplets are suggested to be a sufficient factor of drug response, however, with minimal information on their lipid and protein content, and their changes upon conventional CML treatment.
Table 1 (refs 71-73) Pioglitazone, an antidiabetic drug, was earlier proposed as an adjuvant in some solid tumors (publications from 2000’s) acting via peroxisomes, and, therefore, its action may be less specific than direct effect on fatty acid metabolism.
Line 190. Depletion of cholesterol ethers (CE) should be considered a key event in metabolic reprogramming and malignant cell survival, since cholesterol is a key component of membrane lipid bilayer in normal and malignant cells, and, in simple words, its deficiency is a limiting factor for tumor growth.
Line 227-241: Statins in combined protocols with TKI showed only marginal benefit in terms of complete CML remission rates as recognized by the authors.
Generally, the text is well designed. However, one should denote some basic components of membrane lipids, especially, surface expression of e.c., phosphatidylserine which is playing a key role in the course of apoptosis, the main type of myeloid cell death (spontaneous or upon cancer therapy).
Nevertheless, drawing attention to repurposing of drugs affecting fat metabolism should be stimulatory for new ways to overcome drug resistance in poor responders to TKI. The article would be more compelling if the authors could present: (1) current knowledge on lipid disturbances in less and more differentiated myeloid populations; (2), in vitro effects of cytostatics (or irradiation) on lipid composition of CML cell cultures; (3) results of clinical studies employing statins, PPAR agonists etc. as monotherapy of cancer.
The article is dedicated to quite interesting and intriguing issues of potential repurposing of drugs affecting lipid metabolism for adjuvant treatment in TKI-resistant cases.
Author Response
We thank this reviewer for their time in reviewing our paper and providing valuable feedback for the manuscript. We have addressed the reviewer's comments and suggestions below:
Comment 1: The manuscript concerns alterations of lipid contents and metabolism in chronic myeloid leukemia (CML) and effects of chemotherapy on this metabolic compartment. Available literature is relative scarce, and, in current review, the focus is made on the intracellular lipid accumulation as a factor of drug resistance in some patients who failed to respond to the TKI therapy. Of course, this therapeutic approach may be potentially useful, especially, seeking for repurposing of some drugs affecting lipid metabolism, for supplementary cancer treatment in TKI-resistant cases.
Response 1: We thank the reviewer for recognising the usefulness of targeting lipid storage as a potential therapeutic approach in some patients who fail to respond to TKI therapy. We acknowledge that there is limited available literature on this topic but have critically evaluated the available literature and extrapolated from the results of studies that have indirectly targeted lipid storage in CML. This led to us highlighting the gaps in the literature and suggesting the most suitable direction for future research on lipid storage in CML with a focus drug screening libraries and studies of xenograft models.
Comment 2: Lines 81-84. Before discussing the data on lipid storage as a drug-resistance factor, one should mention in more details some other well-known mechanisms of TKI resistance, i.e., T355I and other mutations of BCR/ABL which make the cells insensitive to TK inhibition.
Response 2: This section of text is now lines 98 - 103. We have now specifically mentioned the T315I mutation as one of the well characterised ABL1 kinase domain mutations. We have also expanded on this discussion adding that switching to an alternative TKI is often sufficient for disease management in patients with BCR::ABL1 resistance mutations, referencing the European Leukaemia Network 2025 recommendations for treating CML authored by Apperley et al..
Comment 3: Line 132. Section 3 lipid droplets are suggested to be a sufficient factor of drug response, however, with minimal information on their lipid and protein content, and their changes upon conventional CML treatment.
Response 3: We thank the reviewer for this comment in highlighting that at this point of the manuscript there was minimal information to justify our exploration of lipid droplets. Now located at line 155 we have modified this sentence to clarify that in the main body text of this review (following this paragraph) we will be investigating the potential relevance of lipid storage in CML. We aim here to provide a transition for the reader into the next chapter where we will further explore evidence of lipid storage being implicated in the TKI response of CML cells. Furthermore, there is limited information available in the literature on how TKI treatment alone effects the lipid and lipid metabolism-protein content in CML. Therefore, we cannot present more detailed information on CML cell lipid content is changed upon conventional CML treatment. Instead, we have presented information reported on changes in lipid and lipid metabolism-protein content in CML cells based on patient treatment response (lines 148-150 & 347-350) and cell stemness (lines 121-124 & 143-144). We also discuss studies on the effect of BCR::ABL1 signalling on lipid content (lines 221-225) and in turn, lipid content-mediated changes in BCR::ABL1 expression (lines 194-204).
Comment 4: Table 1 (refs 71-73) Pioglitazone, an antidiabetic drug, was earlier proposed as an adjuvant in some solid tumors (publications from 2000’s) acting via peroxisomes, and, therefore, its action may be less specific than direct effect on fatty acid metabolism.
Response 4: We acknowledge the necessary addition of general background information on pioglitazone and its use in Type 2 Diabetes (T2D). We have included this information on lines 366-372 since this section of the text felt most appropriate for this information. We reference Waugh et al. when briefly mentioning the insulin-sensitising activity of pioglitazone. We have also updated our sentence located at lines 363-367 regarding the fact that in CML studies that have employed PPARγ agonists, the focus was not on the specific LD-regulatory effects of PPARγ.
Comment 5: Line 190. Depletion of cholesterol ethers (CE) should be considered a key event in metabolic reprogramming and malignant cell survival, since cholesterol is a key component of membrane lipid bilayer in normal and malignant cells, and, in simple words, its deficiency is a limiting factor for tumor growth.
Response 5: We thank the reviewer for highlighting the need to briefly mention the critical function of cholesterol in cells and how it has the potential to be utilised by cancer cells to support their proliferation and survival. This section has been updated to include this information referencing the Maxfield et al. review on the role of cholesterol and is now located at lines 216-219.
Comment 6: Line 227-241: Statins in combined protocols with TKI showed only marginal benefit in terms of complete CML remission rates as recognized by the authors.
Response 6: While the benefit of statin-TKI combinations were marginal, increasing the incidence of DMR can ultimately allow for more patients to trial TFR since sustained DMR is an eligibility requirement for trailing TKI cessation. We have added an additional sentence at lines 279-283 to emphasise this point referencing the ELN 2025 recommendations for CML, authored by Apperley et al..
Comment 7: Generally, the text is well designed. However, one should denote some basic components of membrane lipids, especially, surface expression of e.c., phosphatidylserine which is playing a key role in the course of apoptosis, the main type of myeloid cell death (spontaneous or upon cancer therapy).
Response 7: We thank the reviewer for this suggestion and have updated the section of manuscript that covers the general function of lipids in the cell to include apoptosis as one of the critical functions of lipids. This change is located on line 125.
Comment 8: Nevertheless, drawing attention to repurposing of drugs affecting fat metabolism should be stimulatory for new ways to overcome drug resistance in poor responders to TKI. The article would be more compelling if the authors could present: (1) current knowledge on lipid disturbances in less and more differentiated myeloid populations; (2), in vitro effects of cytostatics (or irradiation) on lipid composition of CML cell cultures; (3) results of clinical studies employing statins, PPAR agonists etc. as monotherapy of cancer.
Response 8: We thank the reviewer for these suggestions to improve our article.
1) There is limited information about lipid specific metabolic changes in different myeloid populations, however we did find that Kinder et al. et al. previously demonstrated that fatty acid lipoxygenase enzyme ALOX15 plays a critical role in the maintenance of haematopoietic stem cells. This suggests and important role of lipid metabolic processes in haematopoiesis and is an important finding to include in our paper when discussing the general cellular functions of lipid metabolism and transitioning in to detailing blood cell specific functions. This update in the text is located at lines 125-127.
2) We did not identify relevant studies employing therapeutics or radiation that did not induce cell death in CML cultures and additionally focussed on the lipid altering effects of these substances. We do however discuss K562 cell response to statins which do not induce cell death at clinically relevant doses, however these studies exceeded the clinical dose range. In vitro studies of statins in K562 cells are located at lines 244-251. These lines also include alterations made to address suggestion 3.
3) To our knowledge, there are no clinical studies of statin or pioglitazone monotherapy as an anti-cancer treatment. This is likely because these drugs do not exhibit cytotoxicity within the clinical dose range. We had mentioned an in vivo murine study of simvastatin monotherapy on K562 xenograft tumour development which successfully demonstrated inhibition of tumour growth. However, to provide further information about the effects of statin monotherapy we have expanded this section to include an additional reference that showed atorvastatin monotherapy was sufficient to reduce cell proliferation through inducing cell cycle arrest in K526 cells. This update is located on lines 244- 252.
Comment 9: The article is dedicated to quite interesting and intriguing issues of potential repurposing of drugs affecting lipid metabolism for adjuvant treatment in TKI-resistant cases.
Response 9: We thank the reviewer for recognising that this article is dedicated to evaluating an interesting, but relatively unexplored potential mechanism of TKI-resistance in CML. We hope that publication of this article will prompt further investigation into lipid metabolism as an adjuvant therapeutic target in combination with TKIs.
Reviewer 2 Report
Comments and Suggestions for Authors
The paper summarizes the current progress in overcoming TKI resistance in CML by targeting lipid metabolism. The review covers the available in vitro and clinical studies that deal with altering lipid metabolism in CML cells at various points: targeting lipid droplets, inhibition of synthesis, targeting of associated proteins, targeting lipophagic regulation. The authors have concluded that, although preclinical studies point to the correlation between lipid metabolism and TKI resistance, the understanding of mechanisms beyond and success in clinical studies are still lacking. They have proposed future directions in research based on detailed investigation of potential biomarkers, drug repurposing, high throughput data processing and in vivo xenograft models.
The manuscript is interesting and well written, and contains useful data for future researh in the field. Minor suggestions:
1. Please consider introducing the list of abbreviations, since there is a large number of them, so the reading is sometimes difficult.
2. Please consider introducing a summary table such as Table 1 for each subchapter (I. e. each target point in lipid metabolism). This will improve visibility of data and help better distinguishing of analyzed studies by type and conclusion.
3. Please add a reference for BioRender software
Author Response
We thank this reviewer for their time in reviewing our paper and providing valuable feedback for the manuscript. We have addressed the reviewer's comments and suggestions below:
Comment 1: The paper summarizes the current progress in overcoming TKI resistance in CML by targeting lipid metabolism. The review covers the available in vitro and clinical studies that deal with altering lipid metabolism in CML cells at various points: targeting lipid droplets, inhibition of synthesis, targeting of associated proteins, targeting lipophagic regulation. The authors have concluded that, although preclinical studies point to the correlation between lipid metabolism and TKI resistance, the understanding of mechanisms beyond and success in clinical studies are still lacking. They have proposed future directions in research based on detailed investigation of potential biomarkers, drug repurposing, high throughput data processing and in vivo xenograft models.
Response 1: We thank the reviewer for their critical insights on this manuscript. We have aimed to review the existing literature focused on both the direct and indirect targeting of lipid storage in CML, with the aim of guiding future research directions.
Comment 2: The manuscript is interesting and well written, and contains useful data for future research in the field. Minor suggestions:
Response 2: We thank the reviewer for their interest in this work and appreciating its contribution to the field.
Comment 3: Please consider introducing the list of abbreviations, since there is a large number of them, so the reading is sometimes difficult.
Response 3: We thank the reviewer for this suggestion to help improve the ease of which readers follow this article. We have now included a list of abbreviations mentioned in the text at lines 43-56.
Comment 4: Please consider introducing a summary table such as Table 1 for each subchapter (I. e. each target point in lipid metabolism). This will improve visibility of data and help better distinguishing of analyzed studies by type and conclusion.
Response 4: We appreciate the reviewer’s suggestion here to improve visibility of the data presented. The studies that serve as the main point of focus for each subchapter are the clinical trials/retrospective studies that are already summarised in Table 1. We believe that the tables for each sub chapter would include a large portion of the same information as that in Table 1 and would potentially appear as repetitive. For this reason, we have not introduced a summary table for each subchapter but hope that the additional background information that we have included on the cellular functions of the lipid pathways discussed in this article (e.g cholesterol in lines 216-219) will more easily distinguish the lipid metabolism target of each section.
Comment 5: Please add a reference for BioRender software
Response 5: The Biorender reference for in each figure legend has been updated from ‘Created in BioRender’ to ‘Figure created using BioRender.com’. Additionally, in the acknowledgements section of the paper we have acknowledged the use of BioRender to prepare the figures for this paper with a link to the BioRender website at lines 571-572.
Round 2
Reviewer 1 Report
Comments and Suggestions for Authors
The article is now supplied with some details on lipid biochemistry in CML and pharmacology of lipid-modifying drugs which are necessary for understanding the main points and pre-requisites of this review. The questions of reviewer are answered, and appropriate data are added to the text. Many proofs and suggestions in this field are still indirect, but the mechanisitic effects and the views of the authors are now more clear. The article will be useful for cell biologists and pharmacologists.